Fluctuating selection across years and phenotypic variation in food-deceptive orchids

Scopece Giovanni giovanni.scopece@unina.it 1
Juillet Nicolas 2
Lexer Christian 3
Cozzolino Salvatore 1
1 Department of Biology, University of Naples Federico II , Naples , Italy
2 UMR Peuplements Végétaux et Bioagresseurs en Milieu Tropical, Université de la Réunion, Pôle de Protection des Plantes , Saint Pierre , La Réunion , France
3 Department of Botany and Biodiversity Research, University of Vienna , Vienna , Austria
Roberts David
Electronic publication date: 2017 Aug 25
Publication date: 2017
Volume: 5
Electronic Location ID: e3704
Received 2017 May 23; Accepted 2017 Jul 27
Copyright: ©2017 Scopece et al.
Copyright year: 2017
Copyright holder: Scopece et al.
License: This is an open access article distributed under the terms of the Creative Commons Attribution License, which permits unrestricted use, distribution, reproduction and adaptation in any medium and for any purpose provided that it is properly attributed. For attribution, the original author(s), title, publication source (PeerJ) and either DOI or URL of the article must be cited.
License URL: https://creativecommons.org/licenses/by/4.0/

Keywords: Food-deceptive orchids, Selection differentials, Floral traits, Fluctuating selection, Orchis mascula, Orchis pauciflora, Phenotypic variation

Funding: UniNA Compagnia di San Paolo This research was carried out in the frame of Programme STAR, financially supported by UniNA and Compagnia di San Paolo. The funders had no role in study design, data collection and analysis, decision to publish, or preparation of the manuscript.

==============================
Nectarless flowers that deceive pollinators offer an opportunity to study asymmetric plant-insect interactions. Orchids are a widely used model for studying these interactions because they encompass several thousand species adopting deceptive pollination systems. High levels of intra-specific phenotypic variation have been reported in deceptive orchids, suggesting a reduced consistency of pollinator-mediated selection on their floral traits. Nevertheless, several studies report on widespread directional selection mediated by pollinators even in these deceptive orchids. In this study we test the hypothesis that the observed selection can fluctuate across years in strength and direction thus likely contributing to the phenotypic variability of this orchid group. We performed a three-year study estimating selection differentials and selection gradients for nine phenotypic traits involved in insect attraction in two Mediterranean orchid species, namely Orchis mascula and O. pauciflora, both relying on a well-described food-deceptive pollination strategy. We found weak directional selection and marginally significant selection gradients in the two investigated species with significant intra-specific differences in selection differentials across years. Our data do not link this variation with a specific environmental cause, but our results suggest that pollinator-mediated selection in food-deceptive orchids can change in strength and in direction over time. In perennial plants, such as orchids, different selection differentials in the same populations in different flowering seasons can contribute to the maintenance of phenotypic variation often reported in deceptive orchids.

Introduction

Pollinator-mediated selection is one of the main processes driving the evolution of floral traits in entomophilous pollen-limited plant species (Fenster et al., 2004). The typical form of entomophilous pollination is based on a reciprocal advantage: the flower offers a reward (typically pollen or nectar) to the pollinator, which transports the pollen to conspecific individuals ensuring reproduction. However, pollinators are a priori to visitation, able to directly assess the amount of nectar reward contained in the flowers and thus depend on the information provided by advertising floral traits (Benitez-Vieyra et al., 2010). Therefore, nectar-producing plants have an advantage in being easily recognisable by pollinators so that once a rewarding flower type is discovered, pollinators usually concentrate on a single species (flower constancy; Waser, 1986). In these plant–pollinator relationships, plants are continuously exposed to the choice of pollinators, which imposes a selection on flower traits (Schiestl & Johnson, 2013). Pollinator-mediated selection is thus considered the main factor shaping changes in phenotypic floral trait distributions (Fenster et al., 2004). Accordingly, the development of easily applicable methods to estimate selection differentials in natural populations (Lande & Arnold, 1983) resulted in the common discovery of directional selection on floral traits in many plant systems (Kingsolver et al., 2001; Hereford, Hansen & Houle, 2004).

Flower constancy is an important prerequisite for the establishment of strong selection (stabilising or directional); when insects concentrate on phenotypes that they can associate with a reward, they favour individuals that are similar, and potentially shape the displacement of the phenotypic distribution of floral traits via the elimination of abnormal phenotypes that are not preferred by the pollinators (Waser, 1986). However, plant–pollinator interactions can be more intricate, and patterns of pollinator-mediated selection may differ in the case of temporarily or spatially asymmetrical and non-constant relationships.

Orchids are widely used as a model to study asymmetric plant-insect interactions because they include many species with nectarless flowers that deceive their pollinators (Ackerman, 1986; Schiestl, 2005; Jersàkovà, Johnson & Kindlman, 2006). Among orchid deceptive pollination strategies, the most common is based on a generalised mimicry of floral traits that pollinators associate with the presence of nectar (generalised food-deception, Dafni, 1984). In these deceptively pollinated species, plant–pollinator interactions do not follow flower constancy behaviour because many species of insects learn to avoid deceptive flowers after a few rewardless visits (Gumbert, 2000; Smithson & Gigord, 2003). This pollinator behaviour can result in disruptive selection leading to high phenotypic variability or, alternatively, to a relaxed selection on floral traits (Juillet & Scopece, 2010) of deceptive orchids. This is also suggested by the high intra-specific phenotypic variability in flower traits that has been observed in this plant group, including colour (Nilsson, 1980), shape and size (Ackerman & Galarza-Pérez, 1991), and fragrance (Moya & Ackerman, 1993; Salzmann et al., 2007). In a survey, Ackerman, Cuevas & Hof (2011) confirmed that such variation is more common in deceptive compared to nectar-rewarding species, thus suggesting that the maintenance of a high phenotypic variability may be linked with the adoption of a deceptive pollination system (see also Salzmann et al., 2007).

Factors maintaining high levels of phenotypic variation in deceptive systems have been investigated in several studies (see Juillet & Scopece, 2010 and references therein), particularly after Heinrich’s (1975) proposal that phenotypic variability decreases pollinator avoidance learning, thereby increasing orchid reproductive success. This hypothesis has also been recently suggested by Stejskal et al. (2015) to explain variation in the labellum patterns of a sexually-deceptive orchid. However, Juillet & Scopece (2010) showed that all attempts to identify a reproductive advantage linked with phenotypic variation in food-deceptive species were non-significant or indicated a lower reproductive success thus underlining the need of more studies to support or discard this hypothesis. Other potential causes that could account for high phenotypic variation in deceptive species are negative frequency dependent selection (e.g., Gigord, Macnair & Smithson, 2001) or genetic drift (Knapp & Rice, 1998; Tremblay et al., 2005; Holderegger, Kamm & Gugerli, 2006; Lawton-Rauh, 2008). However, negative frequency dependent selection was contradicted by several studies that excluded a reproductive advantage of the rarer phenotype (Juillet & Scopece, 2010), both in species with discrete phenotypic polymorphism (Ackerman & Carromero, 2005; Pellegrino et al., 2005; Smithson et al., 2007; Tremblay & Ackerman, 2007) and in species with continuous phenotypic variation (Salzmann et al., 2007; Aragón & Ackerman, 2004).

Nevertheless, despite high levels of phenotypic variation, there is increasing evidence for directional selection on floral traits in food-deceptive orchids. For example, pollination success was found to be correlated with plant height in Cypripedium acaule (O’Connell & Johnston, 1998), to the number of flowers in Anacamptis morio (Johnson & Nilsson, 1999), to spur length in the Disa draconis species complex (Johnson & Steiner, 1997) and in hybrid zones between Anacamptis morio and A. longicornu (Zitari et al., 2012), to flowering time in some deceptive orchids (Sabat & Ackerman, 1996; O’Connell & Johnston, 1998; Sun et al., 2009; but see Sletvold, Grindeland & Ågren, 2010), to plant height, flower number and spur length in Dactylorhiza lapponica (Sletvold, Grindeland & Ågren, 2010) and to flower brightness and contrast in Anacamptis morio (Sletvold et al., 2016). This evidence is unexpected, considering the high levels of phenotypic variation seen in deceptive orchids and suggests that, even in this plant group, directional selection mediated by pollinators may be widespread and strong (but see Cintrón Berdecía & Tremblay, 2003). However, these studies were performed in single flowering seasons and disregarded the fact that across years natural selection can vary in both the strength and direction (Darwin, 1859; Grant & Grant, 1989; Benitez-Vieyra et al., 2012; Sletvold & Ågren, 2014). Temporal variation in patterns of natural selection has been often documented in plants (Harder & Johnson, 2009) and has been linked to variation in the pollinator community (Conner et al., 2003), to the presence of herbivores (Sandring et al., 2007), and to abiotic factors (Maad, 2000; Caruso, Peterson & Ridley, 2003; Cintrón Berdecía & Tremblay, 2003; Maad & Alexandersson, 2004). Despite this evidence, however, the incidence of seasonal variation on selection patterns in deceptive orchid species has been rarely investigated (Tremblay, Ackerman & Pérez, 2010).

By estimating the covariance of pollination success with nine different phenotypic traits, we estimated selection differentials and gradients in two Mediterranean food-deceptive orchid species, Orchis mascula and Orchis pauciflora. In particular, we estimated the strength and direction of natural selection over three consecutive years in a sympatric population of these two species with the aim of specifically understanding whether, in the same population, pollinator-mediated selection shows a concordant pattern over different years. We used two orchid species with similar flower morphology and a common set of pollinators (Van Der Cingel, 1995; Cozzolino et al., 2006; Nilsson, 2008; Valterovà et al., 2007) as replicates to increase the power of our conclusions for Mediterranean food-deceptive orchids.

Materials and Methods

Study system

Orchis mascula and O. pauciflora are closely related species in the orchid subtribe Orchidinae (Aceto et al., 1999). O. mascula is a widespread European species, ranging from Sweden to the northern borders of the Mediterranean basin (Sundermann, 1980). It is typically found in sunny meadows or calcareous grasslands up to 2,400 m in altitude. O. pauciflora is generally found on poorer calcareous soils in the south-eastern and central part of the Mediterranean basin, up to 1,500 m in altitude.

The two species are self compatible but non autogamous and rely on generalised food-deception for pollinator attraction (Van Der Cingel, 1995). Hymenopterans are the most common pollinators of these two species (specifically Bombus sp., but also species of the genus Psithyrus, Eucera, Andrena, Osmia, Anthophora), and reproductive success is severely pollen-limited (e.g., Cozzolino et al., 2006). Clonal propagation is extremely rare in both species.

Our study was performed during Spring of 2002, 2003 and 2004 in sympatric natural populations of O. mascula and O. pauciflora located in the Cilento and Vallo di Diano National Park (Southern Italy). For each species, and in each year, plants were randomly selected in subgroups in an area of approximately 4 km2 where distribution was nearly continuous. We labelled and measured a total of 1,188 individuals (492 O. mascula and 696 O. pauciflora; for details, see the Supplementary Information). Due to the big dimensions of the populations, the likelihood of resampling the same individuals across years was very low.

Morphological measurement and data collection

To investigate pollinator-mediated selection, we measured nine morphological traits that are potentially important for pollinator visual attraction. Morphological trait measurements were obtained on the same day and at a time when all the examined plants were at peak flowering, i.e., when all flowers of the inflorescence were open. For each individual that was sampled in this study, we recorded two “inflorescence size” variables: (1) flower number and (2) plant height (from ground to uppermost flower, to the nearest cm). We measured also three “flower size” variables: (3) labellum width (distance between the edges of the two lateral lobes, to the nearest 0.1 mm), (4) labellum length (distance between the labellum tip and spur mouth, to the nearest 0.1 mm) and (5) spur length (distance between the spur mouth and the spur tip, to the nearest 0.1 mm). To obtain these measurements, sampled flowers were dissected and floral elements were placed between two transparent plastic film sheets (Fig. S1). These sheets were subsequently scanned to obtain digital images in a 300 dpi TIFF format with a coordinate millimetre paper on the back for reference; measures of floral traits were later obtained using ImageJ 1.33 software (Rasband, National Institutes of Health, Bethesda, MD, USA). Finally, as previously described by Bradshaw et al. (1998), we measured the pigment content in flower elements using a spectrophotometric method. Anthocyanin concentration (purple pigment) was estimated from tepals and labellum, extracting the pigment with 0.5-ml methanol/0.1% HCl, and determining the absorbance at 510 nm; carotenoid concentration (yellow pigment) was estimated similarly, using methylene chloride for pigment extraction and measuring absorbance at 450 nm. We thus estimated four “flower colour” variables: (6) anthocyanin content in tepals, (7) anthocyanin content in labellum, (8) carotenoid content in tepals and (9) carotenoid content in labellum.

We estimated pollen limitation (PL) as 1-(mean female fitness of open-pollinated plants/mean female fitness of hand pollinated plants), ranging from 0 to 1. Female fitness of open pollinated plants was recorded on the same individuals used for traits measurements and was defined as the number of fruits produced by an individual relative to its number of flowers; female fitness of hand pollinated plants was calculated similarly based on literature data reporting results of crossing experiments conducted in the same populations (Scopece et al., 2007).

Data analysis

All analyses were performed independently for the two species and were conducted using R 2.1.1 software (R Core Team, 2014). To explore the relationship among floral traits, we performed Spearman’s rank correlation. We examined significant differences in floral traits across years using Kruskal-Wallis tests of difference and post-hoc pairwise comparisons using Nemenyi test. To explore the level of variation of floral traits, for both investigated species, we calculated a coefficient of variation (CV) as the ratio between Standard deviation and mean. CVs were calculated for each trait in the three investigated years and then averaged to obtain a value for each of the two species.

We estimated selection differentials (Lande & Arnold, 1983; Brodie, Moore & Janzen, 1995) as the covariance between relative reproductive success and each morphological trait. Plant reproductive success was measured at the end of the flowering period as fruit set, i.e., number of fruit/number of flower, ranging from 0 to 1. The relative reproductive success of an individual was defined as its fruit set divided by the mean population fruit set. Morphological traits were scaled by population mean and standard deviation (z-scores, Lynch & Walsh, 1998). To compare selection differentials over different years, we computed bootstrap bias-corrected confidence intervals (CIs: Maad & Alexandersson, 2004). Non-overlapping CIs indicated significantly different selection differentials, CIs including 0 were deemed be non-significant.

To estimate selection gradients, following Lande & Arnold (1983), we performed multiple single linear regressions using relative fitness as dependent variable and standardized traits (z-scores) as predictors. This method was used to produce a simple comparison among traits and years, In alternative to a detailed modelling of selection gradients (as performed in Schluter, 1988; Tremblay, 2011) here we used scaled traits (z-scores) to satisfy as best as possible the mathematical assumptions underlying regression equations (i.e., normality).

Results

All floral traits were moderately and positively correlated with the exception of carotenoid content in tepals and labellum, which were negatively correlated with other phenotypic traits (Table 1). In both species, several floral traits showed significant differences across different years (Figs. 1 and 2).

Table 1 Phenotypic correlations (Spearman’s rank) among morphological traits in Orchis mascula (above diagonal) and O. pauciflora (below diagonal).

All plants from the three years were pooled together.

	Plant height	Flower number	Labellum width	Labellum length	Spur length	Anthocyanins content in tepals	Anthocyanins content in labellum	Carotenoid content in tepals	Carotenoid content in labellum	
Plant height		0.76***	0.47***	0.48***	0.44***	0.29***	0.25***	0.07	−0.07	
Flower number	0.51***		0.25***	0.30***	0.24***	0.15**	0.14**	0.15**	0.06	
Lebellum width	0.32***	0.28***		0.71***	0.69***	0.39***	0.48***	−0.16**	−0.09	
Labellum length	0.26***	0.22***	0.49***		0.63***	0.40***	0.50***	0.01	0.06	
Spur length	0.22***	0.10*	0.44***	0.14**		0.47***	0.50***	−0.05	−0.05	
Anthocyanins content in tepals	0.16**	0.19***	0.08	0.22***	0.01		0.66***	−0.01	−0.08	
Anthocyanins content in labellum	0.18***	0.42***	0.31***	0.24***	0.08	0.45		−0.01	0.01	
Carotenoid content in tepals	−0.09*	0.21***	0.19***	−0.03	−0.05	−0.05	0.18***		0.43***	
Carotenoid content in labellum	0.24***	0.19***	0.30***	0.10*	0.00	0.18***	0.25***	0.32***		
Notes.

*** P < 0.001.

** P < 0.01.

* P < 0.05.

Figure 1 Morphological traits (white bars) and selection differentials (grey bars) in Orchis mascula.

(A) Plant height, (B) flower number, (C) labellum width, (D) labellum length, (E) spur length, (F) anthocyanin (tepals), (G) anthocyanin (labellum), (H) carotenoids (tepals), (I) carotenoids (labellum). Different letters indicate significant differences.

Figure 2 Morphological traits (white bars) and selection differentials (grey bars) in Orchis pauciflora.

(A) Plant height, (B) flower number, (C) labellum width, (D) labellum length, (E) spur length, (F) anthocyanin (tepals), (G) anthocyanin (labellum), (H) carotenoids (tepals), (I) carotenoids (labellum). Different letters indicate significant differences.

In both investigated species, CVs were extremely high for all floral traits indicating a high variability of floral traits with a slight variation across years. Average CV was 0.35 in O. mascula and 0.42 in O. pauciflora (See the Supplemental Information).

Data from Scopece et al. (2007) showed that, in both species, the ratio between hand-pollinated flowers and fruits developed was very high (1 for O. mascula and 0.92 for O. pauciflora), in contrast with fruit formation in open-pollinated individuals, which was low (0.11 in O. mascula and 0.10 in O. pauciflora). Thus, both species were severely pollen-limited (PL for O. mascula = 0.89; PL for O. pauciflora = 0.87).

All of the investigated floral traits in both species showed some amount of selection across years (Figs. 1 and 2). A comparison of the selection differentials in three consecutive years revealed significant differences for seven traits in both O. mascula (Fig. 1) and O. pauciflora (Fig. 2). Specifically, in O. mascula, selection differentials were significantly different for six out of nine traits between 2002 and 2003, three out of nine between 2003 and 2004, and three out of nine between 2002 and 2004 (Fig. 1). In O. pauciflora, selection differentials were significantly different for two out of nine traits between 2002 and 2003, six out of nine between 2003 and 2004, and two out of nine between 2002 and 2004 (Fig. 2).

Across the three years, the selection differential patterns were generally consistent with floral trait variation (i.e., decrease when floral traits increase). In O. mascula, the patterns were consistent in seven out of nine traits. In O. pauciflora, the patterns were consistent in six out of nine traits.

We also found changes in the direction of selection (i.e., sign of selection differential) in three out of nine floral traits in O. mascula (Fig. 1) and in three out of nine floral traits in O. pauciflora (Fig. 2). Selection gradients for each investigated population are reported in Table 2. Results suggest only few marginally significant values indicating weak direct selection on some of the floral traits.

Table 2 Selection gradients (ß) calculated according to Lande & Arnold (1983).

	Orchis mascula	Orchis pauciflora	
	2002	2003	2004	2002	2003	2004	
Plant height	−0.203	0.173	0.196	0.281	0.135	0.414**	
Flower number	0.440*	−0.119	−0.063	0.030	−0.132	−0.122	
Labellum width	−0.231	0.018	−0.258*	−0.040	0.017	0.295*	
Labellum length	0.462**	0.070	0.538***	0.005	0.101	−0.411***	
Spur length	−0.217	0.065	−0.076	0.095	0.041	−0.101	
Anthocyanin content in tepals	0.172	−0.033	0.150	−0.038	−0.080	0.039	
Anthocyanin content in labellum	0.084	0.028	−0.102	0.184	0.095	0.030	
Carotenoid content in tepals	0.066	0.245**	−0.119	0.030	−0.016	−0.054	
Carotenoid content in labellum	0.149	−0.062	0.021	0.005	−0.018	0.187	
Notes.

Significant estimates are in bold.

*** P < 0.001.

** P < 0.01.

* P < 0.05.

Discussion

The quantification and interpretation of the direction, strength and causes of natural selection have been at the centre of scientific debate since the formulation of Darwin’s theory (Darwin, 1859). Among the main aims of studies on selection is the capture of snapshots and milestones in the process of phenotypic trait evolution mediated by natural selection. Thus far, many studies have shown that some of the characters appear to have some advantage over others which suggest that phenotypic selection can be monitored in different animal or plant organisms (Kingsolver et al., 2001; Hereford, Hansen & Houle, 2004), generating the idea that directional selection is widespread in natural populations. This idea is confirmed by the general stability of phenotypic traits observed in rewarding plant species, but it apparently conflicts with the elevated variation observed in deceptive species (Ackerman, Cuevas & Hof, 2011). In this study, we estimated selection differentials and gradients for nine floral traits in two Mediterranean food-deceptive orchid species, Orchis mascula and O. pauciflora over three consecutive years. Overall, we found some degree of phenotypic selection on all of the investigated traits but a strong variation in direction and intensity over different years (Figs. 1 and 2). As expected in deceptive orchids (Tremblay et al., 2005), we found that the two investigated species showed high levels of pollinator limitation. Although pollen limitation is not predictive of pollinator-mediated selection, in severely pollen-limited species the strength of selection is mainly due to the action of pollinators (Sletvold, Grindeland & Ågren, 2010; Sletvold & Ågren, 2014). Despite increasing evidence showing pollinator-mediated selection in Mediterranean food-deceptive orchids (Johnson & Steiner, 1997; O’Connell & Johnston, 1998; Johnson & Nilsson, 1999; Sun et al., 2009; Sletvold, Grindeland & Ågren, 2010; Zitari et al., 2012; Sletvold et al., 2016), our results showed a weak and variable selection on floral traits when analysed over different, consecutive seasons suggesting the absence of constant directional selection on these species. As already reported in Tremblay, Ackerman & Pérez (2010) and in Ackerman & Galarza-Pérez (1991), the inconsistency in strength and direction can potentially be responsible for the elevated phenotypic variation that we directly assessed in the investigated species through the calculation of coefficients of variation for morphological traits that were on average higher than those reported in previous literature surveys on deceptive orchid species (i.e., 0.35 in O. mascula and 0.42 in O. pauciflora versus an average of 15.2% in the deceptive species and of 11.1% for the rewarding species reported in Ackerman, Cuevas & Hof, 2011). Indeed, we found that selection differentials significantly varied at least once in direction or strength in seven out of nine traits in both O. mascula and O. pauciflora over different years (Figs. 1 and 2) with a total of 12 changes out of 27 comparisons in O. mascula (Fig. 1) and 10 changes out of 27 in O. pauciflora (Fig. 2). In both investigated species, most of the studied floral traits were positively correlated (Table 1), which could in principle mask the action of natural selection in the selection differential analysis (Lande & Arnold, 1983). However, our analysis of selection gradient showed only a few marginally significant results, thus confirming the weak selection observed in selection differentials (see Table 2).

Floral traits were not constant over different years and most of these traits showed significant differences in different reproductive seasons (Figs. 1 and 2). In several cases, selection differentials were consistent with phenotypic trait variation; e.g., in O. mascula, plant height was under stronger selection when plants were less tall (Fig. 1), thus suggesting a weak directional selection over the years. However, for other traits, selection differentials were inconsistent with this phenotypic variation (Fig. 1 and Fig. 2), which suggests that fluctuations of natural selection for these traits are not an artefact but rather reflect the activities of different selective agents in different years.

Significant differences among floral traits in different years, within the same population, may be explained as a consequence of phenotypic plasticity. Indeed, in a variable environment, a single generalist genotype can potentially express a wide range of random phenotypes or show different responses to environmental cues via phenotypic plasticity (Kawecki & Ebert, 2004; Hill & Mulder, 2010; Morales, Ackerman & Tremblay, 2010). In our study, this can be accentuated by the extreme climatic conditions of the 2003 heat wave (Beniston, 2004).

Interestingly, in 2003, measurements showed a quantitative reduction of most floral traits (Figs. 1 and 2). However, despite the contribution of plasticity on the phenotypic expression of the investigated floral traits, their heritability was verified by a comparison of phenotypic expression of traits in hybrid zones (G Scopece, 2017, unpublished data).

Fluctuating selection has been proposed to be common in natural populations (Siepielski, Di Battista & Carlson, 2009; Kimball et al., 2012), but its potential role is still debated particularly due to the dearth of extensive multi-year studies (Tremblay, Ackerman & Pérez, 2010). In our study, with a multi-year dataset, we showed that the change in the direction and intensity of selection on the same floral traits could determine a continuous but fluctuating pressure that favours different individuals in different years. Within the same population, this can potentially result in the maintenance of variable phenotypes. Indeed, although changes in selective pressures in annual plant species may generate a displacement of phenotypic traits, in perennial long-lived plants such as orchids, the individual’s reproductive success is the result of its performance during its lifetime and thus only a selection that is constant over time can generate displacement or stabilisation of the distribution of phenotypic traits. In contrast, fluctuating selection is more likely to explain the phenotypic variation observed in the natural populations of deceptive orchids.

The main source of this fluctuating selection is most likely the action of pollinators. In the nectar-rewarding orchid Gymnadenia conopsea, spatio-temporal variation in interactions with pollinators contributes to among-years and among-populations variation in selection on floral traits but that several traits are also likely to be subject to different selective agents (Sletvold & Ågren, 2010; Chapurlat, Ågren & Sletvold, 2015). In our study, we focused on severely pollen-limited food-deceptive species in which pollinators are likely to be the main selective agents (Tremblay et al., 2005; Sletvold, Grindeland & Ågren, 2010; Sletvold & Ågren, 2014). Furthermore, direct estimation based on the comparison of reproductive performance between open-pollinated and hand-pollinated plants confirmed elevated levels of pollinator limitation for the two investigated species (PL for O. mascula = 0.89; PL for O. pauciflora = 0.87). Pollinator-mediated selection is a complex process that can be affected by a high number of environmental variables. For example, pollinators with different tongue lengths may exert different selection pressures that positively select flowers with shorter or longer nectar spurs (Johnson & Steiner, 1997). However, pollinator community varies over the flowering season and in different years depending on climatic differences that can alter both the phenology of plants and emergence of pollinators (Fisogni et al., 2016). Moreover, for similar reasons, the surrounding plant community may change in different years, thereby influencing local pollinator preference and abundance (Herrera, 1988; Schemske & Horvitz, 1989). The change in pollinators or surrounding plant community composition is particularly crucial for generalist deceptive species such as O. mascula and O. pauciflora, which attract a wide range of available pollinator species, and is the most likely source of the observed changes in selection differentials. Changes in pollinator-mediated selection have been widely documented in plant species, including orchids, and have been attributed to many environmental variables (e.g., Caruso, Peterson & Ridley, 2003; Conner et al., 2003; Gòmez, 2003; Toräng, Ehrlén & Ågren, 2008; Sletvold & Ågren, 2010). However, identifying the source of variation in selection differentials can only be achieved via detailed community ecological studies at the local scale. For instance, similar attempts have previously shown that vegetation height affects the strength of pollinator-mediated selection in the food-deceptive orchid Dactyloriza lapponica (Sletvold, Grindeland & Ågren, 2013), thus suggesting that variation in selection also occurs within the same reproductive season at small geographic scales.

A different mechanism that have been advocated to explain the maintenance of high phenotypic variation in deceptive species is the genetic drift. In this scenario, repeated founder events and or small effective population sizes would generate phenotypic variation (Gentry & Dodson, 1987; Zimmerman & Aide, 1989; Tremblay & Ackerman, 2001). In support of this scenario is the observation that deceptive orchids typically have lower fruit set and higher fruiting failure than nectar-rewarding species (Neiland & Wilcock, 1998; Tremblay et al., 2005) thus resulting in a reduced effective population size that would consequently increase the chances of genetic drift. In this context, if pollinator-mediated selection is weak, then genetic drift could account for high variation. Fluctuating pollinator-mediated selection and genetic drift can thus both contribute to the observed pattern of high phenotypic variation in deceptive orchids and the relative importance of the two processes should be object of future researches.

Future research avenues should also address the basis of the elevated phenotypic variation of food-deceptive species and confirm whether similar variation occurs even at a fine geographic scale. Such studies should include fine scale community ecological investigations, as for instance temporal and local variation in pollinator community, that aim to disentangle the factors affecting variation. Simultaneously, it would also be important to conduct more studies on nectar-rewarding orchids to test for the opposite pattern, i.e., lower variability of selection pressures in space and time.

Supplemental Information

Figure S1 Scanned digital images used for morphological measurements

Scanned digital images of (A) O. pauciflora and (B) O. mascula floral parts. From left to right: spur, labellum.

Click here for additional data file.

Data S1 Raw data

Morphological measurements and reproductive success of investigated individuals.

Click here for additional data file.

Authors thank AM Nardella and S Impagliazzo for help with field-work and the Cilento and Vallo di Diano National Park for permissions and logistic support. The authors also thank Elodie Chapurlat, RL Tremblay and JD Ackerman that revised a previous version of this manuscript.

Additional Information and Declarations

Competing Interests

Author Contributions

Data Availability

The authors declare there are no competing interests.

Giovanni Scopece conceived and designed the experiments, performed the experiments, wrote the paper, prepared figures and/or tables, reviewed drafts of the paper.

Nicolas Juillet conceived and designed the experiments, performed the experiments, analyzed the data, reviewed drafts of the paper.

Christian Lexer conceived and designed the experiments, wrote the paper, reviewed drafts of the paper.

Salvatore Cozzolino conceived and designed the experiments, contributed reagents/materials/analysis tools, reviewed drafts of the paper.

The following information was supplied regarding data availability:

The raw data has been submitted as a Supplemental File.

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
