# Peer review of "Fluctuating selection across years and phenotypic variation in food-deceptive orchids"

_PeerJ, doi:10.7717/peerj.3704_

## Round 0.1 · original submission · Minor Revisions

· Academic Editor

Minor Revisions

This is certainly an interesting paper and a valuable contribution. With a few minor corrections as outlined by the reviewers I believe it would be acceptable for publication. My main issue is that non-parametric and parametric tests are used without providing the results of tests for normality.

[# Staff Note: It is a PeerJ policy that any suggested references should only be added if the authors agree that they are material #]

·

Basic reporting

The manuscript is very well written, using language largely free of jargon. The rationale for the study is solid with most relevant literature referenced, but some critical articles on variation in selection regimes or fitness among years have been overlooked:

Tremblay, R.L, J. D. Ackerman & M.-E. Pérez. 2010. Riding across the selection landscape: fitness consequences of annual variation in reproductive characteristics. Phil. Trans. R. Soc. B 365: 491-498.
This paper is the first to address annual variation in phenotypic selection regimes in orchids (using two deceptive species), and among the first in botanical studies overall.

Morales, M., J.D. Ackerman & R.L. Tremblay. 2010. Morphological flexibility across an environmental gradient in the epiphytic orchid, Tolumnia variegata: complicating patterns of fitness. Bot J Linn Soc 163: 431-446.
This paper is particularly relevant in the Discussion (e.g., line 307). While the paper focuses on phenotypic plasticity in floral traits across light environments and its consequences to fitness, these patterns were complicated by annual variation in rainfall.

Additional references to support statements are requested for other areas of the manuscript (lines 340-347) and suggestions offered in marked up manuscript.

A matter of language (lines 82-84) "... can result in strong selection for high phenotypic variability ...". It is not selection for variability, but disruptive selection that results in greater variation.

A matter of language (throughout the manuscript): "phenotypic polymorphism" is often equated here to mean “high phenotypic variation” so why not say the latter? The potential problem is that polymorphism sometimes (usually?) refers to discrete morphotypes. While some deceptive species do have discrete morphotypes, most do not and high variation in these species is generally continuous.

The structure does conform to PeerJ standards and the figures are relevant. Readability of Table 2 could be improved by highlighting (boldface) the significant selection gradients.

The raw data supplied are clearly laid out.

Experimental design

The design of data collection is straightforward, and the methodology only needs clarification in a few instances (lines 168 and 171). See marked copy of the manuscript for details. Also, the authors use statistics that assume normal distributions but do not make a statement regarding this assumption (see "validity of the findings" section below).

The research question is clear and covers an important issue largely ignored in discussion of natural selection. Namely, abiotic and biotic conditions vary from year to year and one can expect that the strength and direction of selection to vary as well. Nevertheless, in most empirical and theoretic studies, selection trajectories are either assumed to be constant or the long-term perspective is ignored altogether.

This is a rigorous study, which appears to meet high ethical standards.

Validity of the findings

The authors found complicating patterns of fitness among years for some traits in both their species. And this corroborates the findings of Tremblay et al. 2010.

The data are generally robust, and the statistics may be as well. However, the generation of selection differentials and gradients uses methods that assume variables are normally distributed. Data on reproductive success in deception orchids are notoriously non-normal. If this is the case, then the authors should justify their approach, transform the data, or use other approaches, but you need to define what your variables (binomial, poisson, beta, etc.) are in the model.

The conclusions are clear and very much tied to the original questions. The suggestion to look at "nectar-rewarding orchids to test for the opposite pattern, i.e., lower variability of selection pressures in space and time" (lines 361-363) is an interesting suggestion, and I do expect to see that pattern because rewarding plants start with less variation due to stabilizing selection. Plants would still be subjected to the same year-to-year vagaries of environmental and biotic conditions, as would deception plants.

Comments for the author

With some relatively minor changes mentioned above, this manuscript would be a solid contribution to evolutionary biology. See attached marked pdf for other minor comments and corrections.

·

Basic reporting

The manuscript "Fluctuating selection across years and phenotypic polymorphism in food-deceptive orchids", attempts to demonstrate evidence that selection gradients and direction vary in time in two species of orchids. The paper in general shows a literature survey that is supportive of their ideas, however in euro bias while the evidence they show is much more widely distributed. I think they have an opportunity to show that what they observe here is applicable to many other species. This needs to be strengthen.

I addition the main difficulty i perceive with the paper is the assumption that only natural selection is the cause of evolutionary processes and morphological patterns. The lack of giving sufficient recognition that genetic drift is also an evolutionary process and that it is not included in the discussion of how this may lead to morphological heterogeneity.

Experimental design

The experimental design is acceptable. A series of comments and suggestions have been included in the pdf to help clarify points.

However, note that statistical methods have to be improved and the presentation of the figures have to changed to help the reader understand with more ease the results.

Validity of the findings

The interpretation and the outcome are validated and supported from their findings.

The sample size of the research is good, however, you need to show that the distribution of the data meet normality if you are going to use parametric tests (Tukey).

Note that proportion are not normally distributed thus para metric tests should not be used for those analysis.

Conclusion well stated, although need to expand to non-europeans systems. This would make the paper more applicable to a wider audience.

See comments in pdf for suggestions.

Comments for the author

I have added in the pdf, a series a papers where they can consult. In no way, do I say that these need to cited, these are ideas I am aware of and can be useful in reformatting your paper. I think you should spend a bit of time reading the last section of the Tremblay et al 2005. This gives a holistic approach to probable evolutionary processes in orchids, which tried to demystify the idea that all changes in orchids are solely do to natural selection.

Personally i'm happy that others are doing this type of work in orchids, this has been limited and although Darwin has made many suggestions on how these processes may occur, most of the papers on evolution in orchids are about "patterns" not "processes". Darwin gave examples of "patterns" and 100's of phylogenetic papers on orchids have "evolutionary hypothesis of the possible processes" but very few papers try to disentangle the alternative hypothesis suggested doing the type of work you have done here, going to the field and doing measurements and looking to see if there is apparent "selection advantage" for some characters. This is what Darwin was talking about. Please continue!

Reviewer 3 ·

Basic reporting

This novel and useful contribution addresses a question of broad interest to pollination biologists – how do pollinator preferences influence floral traits? When flowers provide nectar, floral constancy and directional selection tend to reduce floral polymorphism. In deceptive/nectarless systems, polymorphism is more common and there are various unresolved hypotheses about this.
Here, the authors test a new hypothesis based on a robust multi-year analysis of directional selection by pollinators on the traits of two nectarless orchid species.

Basic reporting
The paper is well written with appropriate background from the literature. Ideas and examples are concisely explained. The paper flows well between topics and sections. Raw data file can be opened and is straightforward.
Here are some suggestions for the figures:
Figs 1 and 2 – very well executed.
add extra images to Supplementary figure or new figure suggested- line 144 – How extreme is the polymorphic variation you are documenting? A figure could show this well if the variation is easy to see. Add several photos of each of the orchid species to show it. This would also provide some context for supplementary fig 1 – non-orchid researchers may not know what these bits are!
Supplementary figure 1 – add some lines to show where the morphometric measurements were taken. Maybe export a screen shot of the measurements as in ImageJ with grid background?

Experimental design

Experimental design
The long-term data set is a strong feature of this work. Data collection was well-matched to hypothesis and robust design. Methods for analysis are appropriate.

Validity of the findings

Validity of the findings
Conclusions well-supported by results. The authors take care to use caveats and some caution to evaluate the biological relevance of the results within the context of the study species (e.g. mentioning the importance of the perennial life history of orchids and how this might/might not interact with annual fluctuations in directional selection, discussing strength of pollinator selection given pollinator limitation).

Comments for the author

General comments
92-98 reword as: The factors maintaining high levels of phenotypic polymorphism in deceptive systems have been investigated in several studies (see Juillet & Scopece 2010 and references therein), particularly after Heinrich’s (1975) proposal that phenotypic variability decreases pollinator avoidance learning, thereby increasing orchid reproductive success. This hypothesis has also been recently suggested by Stejskal et al. (2015) to explain variation in the labellum patterns of a sexually-deceptive orchid.

303-356 – this is very long paragraph. Break into ~3 to aid reading and better emphasise your main points.

---

## Round 0.2 · accepted · Accept

· Academic Editor

Accept

Many thanks for the careful corrections. I look forward to see it published.